# An Airborne Gravity Gradient Compensation Method Based on Convolutional and Long Short-Term Memory Neural Networks

**DOI:** 10.3390/s25020421

**Published:** 2025-01-12

**Authors:** Shuai Zhou, Changcheng Yang, Yi Cheng, Jian Jiao

**Affiliations:** College of Geoexploration Science and Technology, Jilin University, Changchun 130012, China

**Keywords:** gravity gradient, deep learning, long short-term memory, convolution, post-error compensation

## Abstract

As gravity exploration technology advances, gravity gradient measurement is becoming an increasingly important method for gravity detection. Airborne gravity gradient measurement is widely used in fields such as resource exploration, mineral detection, and oil and gas exploration. However, the motion and attitude changes of the aircraft can significantly affect the measurement results. To reduce the impact of the dynamic environment on the accuracy of gravity gradient measurements, compensation algorithms and techniques have become a research focus. This paper proposes a post-error compensation algorithm using convolutional and long short-term memory neural networks (CNN-LSTMs). By leveraging convolution feature extraction capabilities and considering the temporal dependencies of dynamic measurement parameters with LSTM, the model demonstrates a stronger ability to learn from severely coupled time series data, resulting in a significant improvement in the compensation performance. This method outperforms traditional neural networks’ multi-layer perceptrons (MLPs) in terms of compensation accuracy on both simulated and measured airborne gravity gradient data from Heilongjiang Province.

## 1. Introduction

A gravity gradiometer is a high-precision instrument that measures gravity gradients and is widely used in fields such as geophysical exploration, military reconnaissance, space measurement, and underground resource exploration. Since the early 20th century, gravity gradiometers have evolved from mechanical to electronic and eventually to quantum technology. The concept of the gravity gradient traces back to Newton’s law of universal gravitation, which described the gravitational field, despite the fact that there were no instruments available at the time to accurately measure the gravity gradient. By the end of the 19th century, scientists recognized the importance of spatial variation in the gravitational field for more precise detection of underground materials. Following this realization, advances in mechanics, mathematics, and material sciences enabled the design of precise gravity-measuring instruments [1,2]. Torsion balance-based gravity gradiometers first appeared in the early 20th century. These torsion balance-type instruments calculated the gravity gradient by measuring force differences on suspended masses within a gravitational field. A representative example is the Eötvös torsion balance, developed by Hungarian physicist Eötvös Loránd around 1900, which was used to measure variations in the earth’s gravitational field. This design became a classic instrument in early gravity gradient measurement, laying the foundation for subsequent developments [3,4]. With the advancement in airborne and marine technologies, mechanical gravity gradiometers were gradually applied in these fields by the mid-20th century. Because of the high speed of aircraft and ships, these instruments had to meet higher interference resistance requirements [5]. Early mechanical instruments were bulky and had low sensitivity, with measurement accuracy limited by platform vibrations and environmental factors, allowing only low-speed, localized measurements [6]. Gravity gradiometers were revolutionized in the 1950s due to rapid technological advancements. Instruments built with capacitive sensors, piezoelectric materials, and other electronic components gradually replaced traditional mechanical structures. These electronic instruments measured tiny displacements or torques caused by gravitational fields and used electronic amplifiers to improve signal sensitivity [7,8]. In 1979, the American Air Force introduced the Mark II electronic gravity gradiometer, marking a significant leap in gravity measurement technology. This instrument used high-precision sensors to capture gravity gradient signals aboard aircraft, increasing sensitivity while also allowing measurements on dynamic platforms [9]. During this period, electronic gravity gradiometers were widely used in airborne and marine gravity surveys, and they eventually found use in underground resource exploration. During the 1980s, gravity gradiometers were widely used in marine exploration, particularly for detecting oil, gas, and mineral resources. Because gravity gradients are more sensitive to underground structural changes than gravity values, marine gravity gradient measurements may be more accurate in determining resource distributions. These instruments were typically mounted on deep-sea detectors or ships, capturing small changes in the seafloor’s gravitational field to conduct high-resolution surveys [10,11]. By the late 20th century, superconducting technology had significantly improved the accuracy of gravity gradiometers. Superconducting gravity gradiometers use superconductors’ zero-impedance properties to significantly reduce environmental noise interference, resulting in extreme sensitivity. These instruments were mainly used for scientific research and military detection, which required precise measurements. Superconducting gravity gradiometers operate at low temperatures, making them ideal for precise gravity measurements in polar or space environments [12,13]. In the early 21st century, the emergence of cold atom technology revolutionized gravity gradient measurement. Cold atom technology, which is based on quantum mechanics, cools atoms to near absolute zero, resulting in the ability to measure the gravity gradient through atomic interference. This technology theoretically provides extremely high measurement sensitivity while avoiding the mechanical and thermal noise issues found in traditional instruments. In 2007, Paris-Sud University in France proposed using atomic interferometers to measure gravity gradients. Subsequently, cold atom gravity gradiometers evolved from experimental prototypes to practical devices. Cold atom gravity gradiometers provide more stable and precise measurements under dynamic conditions than traditional electronic gravity gradiometers, making them ideal for applications in aviation, space, and ground exploration [14,15].

As airborne gravity gradiometers improved, researchers gradually realized that the aircraft’s movement and attitude changes could have a significant impact on measurement results. The errors in dynamic airborne gravity gradient measurements can be classified into two major categories: First, there are system errors caused by noise within the gravity gradiometer itself, such as scale factor inconsistencies in accelerometers within rotating accelerometer-based gradiometers, installation errors, circuit gain mismatches, and other system error components that are related to the manufacturing process and system performance. Second, errors from the dynamic measurement environment, such as translational motion, angular motion, attitude, and temperature, are coupled with various error channels in the gradiometer system, transferring noise interference from the dynamic environment to the gravity gradiometer’s output, resulting in dynamic measurement errors in gravity gradient measurements [16]. To reduce the impact of the aircraft’s dynamic environment on the accuracy of gravity gradient measurements, compensation algorithms and technologies have emerged as a key research focus. In the 1970s, researchers proposed using aircraft attitude and velocity information for preliminary compensation. By utilizing motion information provided by the aircraft’s inertial navigation system, initial corrections could be made to compensate for errors caused by acceleration and angular velocity [17]. Gravity gradient compensation technology became more refined in the 21st century as computer technology, navigation systems, and high-precision sensors advanced rapidly. The use of advanced inertial measurement units (IMUs) and global positioning systems (GPSs) enabled the development of precise dynamic compensation techniques. Multisensor data fusion technology were integrated into airborne gravity gradient compensation systems. By integrating data from sources such as IMUs and GPSs, errors caused by aircraft motion and air turbulence can be effectively eliminated, significantly improving the accuracy of gravity gradient measurements [18]. In recent years, the advancement of artificial intelligence and machine learning technologies has provided new approaches to airborne gravity gradient compensation.

Deep learning is a branch of machine learning that has made significant advances in a variety of fields in recent years, owing to increased computational power and the availability of large amounts of data. Deep learning has outperformed traditional methods in several areas, including image recognition, speech processing, and natural language processing. The concept of deep learning originated from neuroscience research in the 1940s when McCulloch and Pitts proposed a neuron model that simulated the basic functions of biological neurons [19]. Rosenblatt introduced the Perceptron model in 1957, which is widely regarded as the first artificial neural network [20]. The perceptron was a simple linear classifier that performed well with basic problems. However, in 1969, Minsky and Papert discovered that the perceptron could not solve nonlinear problems, such as the XOR (exclusive OR) problem, causing a temporary halt in neural networks research [21]. In the 1980s, neural networks regained attention with the introduction of the backpropagation algorithm. Hinton and others popularized backpropagation in 1986, allowing neural networks to effectively adjust weights across multiple layers of neurons, thereby addressing complex nonlinear problems [22]. Though the neural networks of that era were shallow, this progress laid the groundwork for deeper neural networks. Deep learning first emerged in the 21st century as a result of advances in hardware, particularly GPU technology. In 2006, Hinton and colleagues introduced deep belief networks (DBNs), marking the formal rise of deep learning. To address the challenge of training deep neural networks, DBN employed unsupervised layer-wise pretraining [23]. In 2012, Alex Krizhevsky, Ilya Sutskever, and Geoffrey Hinton won the ImageNet competition with their proposal of AlexNet, an eight-layer convolutional neural network. AlexNet’s use of GPUs for training significantly improved computational efficiency [24]. Later neural networks, such as VGGNet (2014) and ResNet (2015), improved neural networks’ depth and performance. ResNet addressed the problem of vanishing gradients in deep networks by introducing residual connections, which enabled networks to reach hundreds of layers [25]. Recurrent Neural Networks (RNNs) are excellent at handling sequential data, such as time series and natural language. However, standard RNNs have difficulty with long-term dependencies and capturing distant information. Hochreiter and Schmidhuber introduced long short-term memory (LSTM) networks in 1997, which combined memory cells and gating mechanisms to effectively address vanishing and exploding gradients. In recent years, LSTM and their simplified variant, GRU (gated recurrent unit), have been widely used in speech recognition, natural language processing, and other fields [26]. In 2017, Vaswani and colleagues introduced the Transformer model, which replaced the RNN structure with a self-attention mechanism for sequential data processing. The Transformer model’s multi-head self-attention mechanism established dependencies between any two positions in a sequence, significantly improving parallel computation efficiency. Transformer-based architectures such as BERTs (bidirectional encoder representations from transformers) and GPT (generative pre-trained transformer) have demonstrated great success in natural language processing tasks, confirming deep learning’s dominance in the NLP (natural language processing) domain [27]. In 2023, Krichen introduced the fundamentals of convolutional neural networks (CNNs), then discussed several popular CNN architectures and compared their performance. He also explored when to use CNNs, their advantages and limitations, and provided recommendations for developers and data scientists [28]. In the same year, Alahmari et al. introduced three CNN models to measure students’ engagement in E-learning (EL) assignments and demonstrated the efficiency under different conditions. The application of these models is primarily based on analyzing learners’ facial expressions in online environments [29].

During flight, an aircraft’s acceleration changes gradually and smoothly over time, rather than abruptly or instantaneously. This behavior is based on fundamental mechanical and aerodynamic principles. Acceleration is the rate at which velocity changes, and an aircraft’s speed is influenced by the interactions of various forces, including thrust, drag, gravity, and lift. These forces usually change continuously during flight, resulting in constant acceleration [30,31,32]. Moreover, as observed from the error transmission mechanism in the rotating gravity gradiometers, there is significant coupling between the measured dynamic parameters and the system errors.

Therefore, considering the temporal correlation of the measured dynamic parameters and their coupling with system errors, this paper employs the CNN-LSTM neural network’s post-error compensation method to achieve compensation for dynamic measurement errors in the rotating gravity gradiometer. First, multi-channel convolution is used to enrich the feature space and improve the model’s feature extraction capability. This process prepares the data so that the subsequent LSTM model can better capture dependencies within input sequences. The features are then passed to the LSTM neural networks, which use a unique cell state and appropriate gating to selectively retain, update, or forget information as required. During backpropagation, the LSTM neural networks can effectively capture temporal dependency information, resulting in superior performance in airborne gravity gradient compensation and improved compensation accuracy. This post-error compensation method effectively suppresses dynamic measurement noise, and its effectiveness was demonstrated using both simulated and measured data.

## 2. Measurement Equations and Error Transmission Mechanism of Rotating Gravity Gradiometer

### 2.1. Measurement Equations of Rotating Gravity Gradiometer

The distance between the test mass centroid of the accelerometer and the center of the disk is denoted as R, and the angle between the line connecting the inspection centroid of the accelerometer to the origin and the positive half of the X-axis is represented as $\theta_{i}$. The disk rotates counterclockwise with a constant angular speed ω. Figure 1 shows how the gravity gradient instrument (GGI) works, with the measurement coordinate system remaining stationary relative to the rotating disk and the inertial coordinate system, and the inertial coordinate system with the origin at point O is chosen as the inertial reference frame.

Based on the working principle of the rotating gravity gradiometer, the output of the gradiometer$\;E_{out}$ can be determined by calculating the accelerometer position vector $R_{i}$, the acceleration $a_{i}$ of the accelerometer under the coordinate system, the acceleration $a_{Ii}$ in the direction of the accelerometer’s input axis, and the accelerometer output $U_{i}$. Let the position *i* vector of the accelerometer be denoted as ${\left(X_{i},Y_{i},Z_{i}\right)}^{T}$(i=1,2,3,4), then the expression of each component of $R_{i}$ is as follows:(1)$$\left\{\begin{array}{l} X_{i}=R\cos\theta_{i}\\ Y_{i}=R\sin\theta_{i}\\ Z_{i}=0 \end{array}\right.$$
where $\theta_{i}=\omega t+\frac{1}{2}(i-1)\pi$, ω represents the angular speed of the GGI, and *t* denotes time.

If the acceleration $a_{0}$ at the origin O is ${\left(a_{X_{0}},a_{Y_{0}},a_{Z_{0}}\right)}^{T}$ and the acceleration $a_{i}$ of the accelerometer *i* is ${\left(a_{X_{i}},a_{Y_{i}},a_{Z_{i}}\right)}^{T}$, then the Taylor first-order expansion of $a_{i}$ at the origin is as follows:(2)$$a_{i}=a_{0}+\Gamma \cdot R_{i}$$
where Γ is the gravity tensor matrix, the unit vector $e_{Ii}$ of accelerometer i in the input axial direction is $\left(\sin\theta_{i},-\cos\theta_{i},0\right)$, and $a_{i}$ represents the acceleration sensitive to the accelerometer that is the projection in the input axis direction, the acceleration $a_{Ii}$ in the input axis direction of accelerometer i is as follows:(3)$$\begin{array}{l} a_{Ii}=a_{i}\cdot e_{Ii}=a_{X_{i}}\sin\theta_{i}-a_{Y_{i}}\cos\theta_{i}\\ \;\;\;\;=a_{X_{0}}\sin\theta_{i}-a_{Y_{0}}\sin\theta_{i}+\\ \;\;\;\;\;\;\frac{1}{2}R(\Gamma_{XX}-\Gamma_{YY})\sin2\theta_{i}-R\Gamma_{XY}\cos2\theta_{i} \end{array}$$

The accelerometer is also sensitive to rotational acceleration in the direction of the input axis. It is generally assumed that the rotational speed of the GGI disk is generally assumed to be constant; thus, the rotational acceleration represents the component of the measurement coordinate system’s variable-speed rotation around the Z-axis in relation to the inertial system. If the acceleration of the rotation angle at the position of the accelerometer inspection centroid is $a_{\omega_{Z}}$, then $a_{Ii}$ becomes(4)$$\begin{array}{l} a_{Ii}=a_{X_{0}}\sin\theta_{i}-a_{Y_{0}}\sin\theta_{i}+\frac{1}{2}R(\Gamma_{XX}-\Gamma_{YY})\sin2\theta_{i}\\ \;\;\;\;\;\;-R\Gamma_{XY}\cos2\theta_{i}-Ra_{\omega_{Z}} \end{array}$$

The output of each accelerometer can be calculated by multiplying the acceleration in the direction of the accelerometer’s input axis by its scale factor. If each accelerometer has the same scale factor, set to k, then the output $U_{i}$ of the accelerometer i is as follows:(5)$$\begin{array}{l} U_{i}=ka_{X_{O}}\sin\theta_{i}-ka_{Y_{O}}\sin\theta_{i}\\ +\frac{1}{2}kR(\Gamma_{XX}-\Gamma_{YY})\sin2\theta_{i}-kR\Gamma_{XY}\cos2\theta_{i}-kRa_{\omega_{Z}} \end{array}$$

If the inspection centroid of accelerometer 1 at time t=0 is located on the positive semi-axis of the x-axis, then $\theta_{i}=\omega t+\frac{\pi }{2}(i-1)$. To eliminate the effect of common-mode acceleration $a_{X_{0}}$, $a_{Y_{0}}$, the output relative to the accelerometer is summed as follows:(6)$$U_{1}+U_{3}=kR(\Gamma_{XX}-\Gamma_{YY})\sin2\omega t-2kR\Gamma_{XY}\cos2\omega t_{i}-2kRa_{\omega_{Z}}$$(7)$$U_{2}+U_{4}=kR(-\Gamma_{XX}+\Gamma_{YY})\sin2\omega t+2kR\Gamma_{XY}\cos2\omega t_{i}-2kRa_{\omega_{Z}}$$

The difference between the sum of the output of the two sets of accelerometers in relative position can remove the influence of the rotational angular acceleration $a_{\omega_{Z}}$, and the gravimeter measurement equation is ideally as follows:(8)$$\begin{array}{ll} E_{out} & =U_{1}+U_{3}-(U_{2}+U_{4})\\ & =2kR(-\Gamma_{XX}+\Gamma_{YY})\times \sin2\omega t-4kR\Gamma_{XY}\cos2\omega t_{i} \end{array}$$

The gravity gradient component is obtained by demodulating the gravity gradiometer’s output signal. This paper uses the digital signal processing method to demodulate the gravity gradiometer output. First, the analog signal output by the gravity gradiometer is discretized into a digital signal using low-pass filtering, and finally, the demodulated $\Gamma_{XX}-\Gamma_{YY}$ component and $\Gamma_{XY}$ components are obtained.

### 2.2. Error Transmission Mechanism of Rotating Gravity Gradiometer

#### 2.2.1. Installation Errors of Accelerometers

Accelerometer installation errors include both positional and sensitive axis errors. Due to mechanical implementation limitations, position installation errors occur when there is a discrepancy between the actual position and the ideal position during its installation on the GGI disk. The installation errors can be described in three parts.

As shown in Figure 2, a rectangular coordinate system is established, with the ideal installation position O of the accelerometer’s centroid serving as the origin. The Z-axis is parallel to GGI’s rotation axis, the X-axis passes through the disk’s center, and the Y-axis is tangential to the rotation trajectory. The error in the actual installation position of the accelerometer’s centroid and the ideal position can be described using errors in three directions: vertical error, radial error, and tangential error, with the positive directions being the coordinate system’s axes. They are denoted as $Z_{i}$, $R_{i}$, and $\gamma_{i}$, respectively. Hence, the position vector of the actual installation location of the accelerometer is expressed as follows:(9)$$\left\{\begin{array}{l} X_{i}^{\prime }=(R+R_{i})\cos\theta_{i}+\gamma_{i}\sin\theta_{i}\\ Y_{i}^{\prime }=(R+R_{i})\cos\theta_{i}-\gamma_{i}\sin\theta_{i}\\ Z_{i}^{\prime }=z_{i} \end{array}\right.$$
where R represents the distance from the center of mass of the accelerometer to the center of the GGI disk and $\theta_{i}$ is the GGI angular velocity of rotation.

When only the vertical error of the inspection centroid is considered, the error term that appears in the measurement equation is as follows:(10)$$\begin{array}{l} \Delta U_{Z}=k(-z_{1}\Gamma_{YZ}-z_{2}\Gamma_{XZ}+z_{3}\Gamma_{YZ}+z_{4}\Gamma_{XZ})\cos\theta \\ \;\;\;\;\;\;\;\;\;\;\;+k(z_{1}\Gamma_{XZ}-z_{2}\Gamma_{YZ}+-z_{3}\Gamma_{XZ}+z_{4}\Gamma_{YZ})\sin\theta \end{array}$$
where *k* represents the accelerometer scale factor, $\Gamma_{YZ}$ and $\Gamma_{XZ}$ are the gravitational gradient components.

When only the radial error of the inspection centroid is considered, the error term that appears in the measurement equation is as follows:(11)$$\begin{array}{l} \Delta U_{R}=\frac{1}{2}k(R_{1}+R_{2}+R_{3}+R_{4})((\Gamma_{XX}-\Gamma_{YY})\sin2\theta \\ \;\;\;\;\;\;\;\;\;\;\;\;\;-\;2\Gamma_{XY}\cos2\theta )-k(R_{1}-R_{2}+R_{3}-R_{4})a_{\omega_{Z}} \end{array}$$
where $a_{\omega_{Z}}$ represents the angular acceleration of the rotation of the accelerometer relative to the Z-axis of the inertial system.

When only the centroid’s tangential error is considered, the error term that appears in the measurement equation is as follows.(12)$$\begin{array}{l} \Delta U_{\gamma }=\frac{1}{2}k(\gamma_{1}-\gamma_{2}+\gamma_{3}-\gamma_{4})(\Gamma_{XX}+\Gamma_{YY}+2\omega^{2})\\ \;\;\;\;\;\;\;\;\;\;\;\;\;-\frac{1}{2}k(\gamma_{1}+\gamma_{2}+\gamma_{3}+\gamma_{4})(\Gamma_{XX}-\Gamma_{YY})\cos2\theta \\ \;\;\;\;\;\;\;\;\;\;\;\;\;-k(\gamma_{1}+\gamma_{2}+\gamma_{3}+\gamma_{4})\Gamma_{XY}\sin2\theta \end{array}$$

The sensitive axis installation error is similar to the position installation error, but its sensitive axis direction deviates from the ideal direction, resulting in the accelerometer-sensitive axis installation error. Assume that the ideal direction of the sensitive axis rotates by angles $\alpha_{Ii}$, $\alpha_{Oi}$, and $\alpha_{Pi}$ around its input axis, output axis, and gimbal axis, respectively, and it is parallel to the sensitive axis’s actual direction in this case. The clockwise direction of rotation is considered positive.

To determine the actual acceleration sensed by the accelerometer, project the accelerations in the input, output, and gimbal axes directions onto the actual input direction.(13)$$\left(\begin{array}{c} a_{Ii}^{\prime }\\ a_{Oi}^{\prime }\\ a_{Pi}^{\prime } \end{array}\right)=\left(\begin{array}{ccc} \cos\alpha_{Pi} & \;\sin\alpha_{Pi} & \;0\\ -\sin\alpha_{Pi} & \;\cos\alpha_{Pi} & \;0\\ \;0 & \;\;\;0 & \;\;\;1 \end{array}\right)\left(\begin{array}{ccc} \cos\alpha_{Oi} & \;0\; & \sin\alpha_{Oi}\;\\ \;0 & \;1 & \;0\\ \;-\sin\alpha_{Oi} & \;0 & \;\cos\alpha_{Oi} \end{array}\right)\left(\begin{array}{ccc} 1 & \;0 & \;0\\ 0 & \;\cos\alpha_{Ii} & \;\sin\alpha_{Ii}\\ 0 & \;-\sin\alpha_{Ii} & \;\cos\alpha_{Ii} \end{array}\right)\left(\begin{array}{c} a_{Ii}\\ a_{Oi}\\ a_{Pi} \end{array}\right)$$

If only the actual sensitive axis is considered to rotate around the input axis, the measurement remains unchanged.

When considering only the rotation of the actual sensitive axis around the output axis, the change in projection of the sensitive axis in the disk plane is minimal and may be negligible. In this case, the measurement equation includes the following error term:(14)$$\begin{array}{l} \Delta U_{O}=ka_{z0}(\tan\alpha_{01}-\tan\alpha_{O2}+\tan\alpha_{O3}-\tan\alpha_{O4})\\ +kR\Gamma_{XZ}((\tan\alpha_{01}-\tan\alpha_{O3})\cos\theta +(\tan\alpha_{02}-\tan\alpha_{O4})\sin\theta )\\ +kR\Gamma_{YZ}((\tan\alpha_{01}-\tan\alpha_{O3})\sin\theta +(-\tan\alpha_{02}+\tan\alpha_{O4})\cos\theta ) \end{array}$$
where $a_{z0}$ represents the acceleration component in the Z-axis direction of the center point of the GGI disk.

When only the actual sensitive axis is assumed to be rotated about the gimbal axis, the error term in the measurement equation is(15)$$\begin{array}{l} \Delta U_{P}=a_{\omega_{Z}}kR(-\cos\alpha_{P1}+\cos\alpha_{P2}-\cos\alpha_{P3}+\cos\alpha_{P4})\\ +a_{XO}k(\sin(\theta +\alpha_{P1})-\cos(\theta +\alpha_{P2})-\sin(\theta +\alpha_{P3})+\cos(\theta +\alpha_{P4}))\\ +a_{YO}k(-\cos(\theta +\alpha_{P1})-\sin(\theta +\alpha_{P2})+\cos(\theta +\alpha_{P3})+\sin(\theta +\alpha_{P4}))\\ +k\omega^{2}R(\sin\alpha_{P1}-\sin\alpha_{P2}+\sin\alpha_{P3}-\sin\alpha_{P4})\\ +0.5kR(\sin\alpha_{P1}-\sin\alpha_{P2}+\sin\alpha_{P3}-\sin\alpha_{P4})(\Gamma_{XX}+\Gamma_{YY})\\ -kR(\cos(2\theta +\alpha_{P1})+\cos(2\theta +\alpha_{P2})+\cos(2\theta +\alpha_{P3})\\ +\cos(2\theta +\alpha_{P4})-4)\Gamma_{XY}+0.5kR(\sin(2\theta +\alpha_{P1})+\sin(2\theta +\alpha_{P2})\\ +\sin(2\theta +\alpha_{P3})+\sin(2\theta +\alpha_{P4})-4)(\Gamma_{XX}-\Gamma_{YY}) \end{array}$$
where $a_{XO}$ and $a_{YO}$ are the translational accelerations in the X and Y directions of the GGI center, respectively.

#### 2.2.2. The Scale Factor Is Inconsistent

Under ideal conditions, we expect the parameters of all accelerometers to be consistent. However, limited by the manufacturing art level, the parameters’ consistency of all accelerometers, such as scale factor, is far from the requirements of application [33]. This inconsistency has an impact on the calculation of the output of the rotating gravity gradiometer because the differential output between the accelerometers cannot eliminate the common-mode quantities generated by translational and rotational motions. Each accelerometer generates the following output:(16)$$U_{i}^{\prime }=k_{i}(a_{X_{0}}\sin\theta_{i}-a_{Y_{0}}\sin\theta_{i}+\frac{1}{2}R(\Gamma_{XX}-\Gamma_{YY})\sin2\theta_{i}-R\Gamma_{XY}\cos2\theta_{i}-Ra_{\omega_{Z}})$$

Differentiating the output of each accelerometer results in the following error term:(17)$$\begin{array}{l} \Delta U_{k}=(a_{X_{0}}(k_{1}-k_{3})-a_{Y_{0}}(k_{2}-k_{4}))\sin\theta \\ \;\;\;\;\;\;\;\;\;-\;(a_{Y_{0}}(k_{1}-k_{3})+a_{X_{0}}(k_{2}-k_{4}))\cos\theta \\ \;\;\;\;\;\;\;\;\;+\frac{1}{2}R(\Gamma_{XX}-\Gamma_{YY})(k_{1}+k_{2}+k_{3}+k_{4}-4k)\sin2\theta \;\;\\ \;\;\;\;\;\;\;\;\;-\;R\Gamma_{XY}(k_{1}+k_{2}+k_{3}+k_{4}-4k)\cos2\theta \\ \;\;\;\;\;\;\;\;\;-(k_{1}-k_{2}+k_{3}-k_{4})Ra_{\omega_{Z}} \end{array}$$
where k is the ideal scaling factor of the accelerometer and $k_{i}$ represents the scale factor of the *i*-th accelerometer.

#### 2.2.3. Residual Angular Motion of the Gradiometer

During gravity gradient measurement, the rotating gravity gradiometer is mounted on a stable platform to minimize the rotation of the measurement coordinate system relative to the inertial frame. However, residual angular motion can still occur, and the accelerometer is sensitive to this aspect of the acceleration component, causing errors. These residual angular motion errors can be divided into two categories: those caused by rotational acceleration and those caused by centrifugal acceleration.

Similarly, residual angular motion influences the derivation of the acceleration vector at the accelerometer’s position, and the acceleration here becomes(18)$$a_{i}^{\prime }=a_{0}+\Gamma \cdot R_{i}-\omega \times (\omega \times R_{i})+a_{\omega }\times R_{i}$$
where $a_{i}^{\prime }$ represents the acceleration vector at the accelerometer, $a_{0}$ is the acceleration vector at the center of the GGI disk, Γ represents the gradient tensor matrix, $R_{i}$ denotes the position vector of the accelerometer, ω is the rotational angular velocity vector at the accelerometer, and $a_{\omega }$ represents the angular acceleration vector of rotation at the accelerometer.

When considering only residual angular motion, the error caused by rotational acceleration is the common-mode error produced by each accelerometer, which will be eliminated in differential computations for the accelerometers. The error due to centrifugal acceleration is expressed as follows:
(19)$$\Delta U_{\omega }=2kR({\omega_{Y}}^{2}-{\omega_{X}}^{2})\sin2\theta +4kR\omega_{X}\omega_{Y}\cos2\theta$$
where *R* represents the distance from the inspection centroid of the accelerometer to the center of the GGI; $\omega_{X}$ and $\omega_{Y}$ are the angular velocities of the measured coordinate system relative to the rotation of the inertial frame around the X and Y axes, respectively.

#### 2.2.4. Accelerometer Nonlinearity Coefficients

During practical measurements, accelerometers are not only linearly sensitive to acceleration along their input axis, as theoretically predicted. In addition to the aforementioned factors, the accelerometer’s output is related to other nonlinear coefficients, and acceleration in the direction of the output and gimbal axes also influences the accelerometer’s output. Since the magnitude of the coefficients beyond the accelerometer’s third order is small, their effect on measurement results is negligible. When only the nonlinear coefficients of the accelerometer are taken into account, the output can be represented as follows:(20)$$\begin{array}{l} U_{i}=k_{i}(K_{Oi}+a_{Ii}+K_{Ii}^{\prime }a_{Ii}^{2}+K_{Oi}^{\prime }a_{Oi}^{2}+K_{Pi}^{\prime }a_{Pi}^{2}\\ \;\;\;\;\;\;\;\;\;\;+K_{IPi}a_{Ii}a_{Pi}+K_{IOi}a_{Ii}a_{Oi}+K_{OPi}a_{Pi}a_{Oi}) \end{array}$$
where $U_{i}$ represents the output of the accelerometer *I*, $a_{Ii}$ is the acceleration in the direction of the input axis, and $a_{Oi}$ denotes the acceleration in the direction of the output axis, ideally in the plane of the GGI disk; $a_{Pi}$ represents the acceleration in the direction of the gimbal axis, ideally in the same direction as the Z-axis of the measurement coordinate system, $k_{i}$ denotes the scale factor, and $K_{Oi}$ represents the zero offset; $K_{IPi}$, $K_{IOi}$, and $K_{OPi}$ are the cross-coupling coefficients; and $K_{Ii}^{\prime }$, $K_{Oi}^{\prime }$, and $K_{Pi}^{\prime }$ are second-order coefficients.

The above errors are discussed individually, but the expressions for some specific errors are still complex. In practice, we need to consider multiple errors comprehensively, leading to extremely complex error expressions. From this, we can understand that the coupling between the measured dynamic parameters and the system errors is highly significant.

## 3. Gravity Gradient Compensation Model

In this paper, a sample set is created by collecting three-directional acceleration, angular velocity, angular acceleration, and gravity gradiometer output noise data obtained during the airborne gravity gradient dynamic measurement procedure. Deep learning neural networks are used to construct a relationship between dynamic measurement errors and motion parameters in order to achieve post-error compensation for gravity gradiometer dynamic measurement data. Given the coupling of noise in airborne gravity gradient data, the proposed neural network model (shown in Figure 3) improves feature extraction capabilities via convolution, which is then passed on to the LSTM neural network. This allows the model to accurately capture temporal dependencies in the data.

In an LSTM neural network, there is a forget gate that can selectively retain or discard previous state information as needed. This allows the LSTM to selectively keep historical information that is useful for the model when handling time series, instead of either discarding or retaining everything indiscriminately. The LSTM controls the inflow of new information through the input gate and manages the output at each time step through the output gate. This gating mechanism allows the model to flexibly adjust the amount of information to update at each moment. In this way, at different points in the sequence, the LSTM can dynamically choose which information to retain and which to output, thereby effectively handling complex temporal dependencies. The cell state in an LSTM serves as a highway running through the entire sequence, making it less susceptible to vanishing gradient issues. This design enables the LSTM to maintain effective state transmission even across long time steps, allowing long-term information to be carried forward to later points in the sequence. This addresses the vanishing gradient problem that traditional RNNs face when learning from long sequences.

The LSTM’s unique structure includes a gate mechanism that controls the flow of information. The hidden layer consists of three gates: the forget gate, the input gate, and the output gate, as well as a cell state, which stores important information. The forget gate uses a sigmoid activation function to generate a value between 0 and 1 based on the current input and the previous hidden state, which determines how much information is forgotten. The closer the value is to 0, the greater the degree of forgetting, and the closer it is to 1, the more information is retained.(21)$$f_{t}=\sigma (W_{f}\cdot [h_{t-1},x_{t}]+b_{f})$$
where $f_{t}$ represents the output of the forget gate, σ is the sigmoid activation function, $W_{f}$ represents the weight matrix of the forget gate, $b_{f}$ denotes the bias, and $\left[h_{t-1},x_{t}\right]$ denotes the combination of the current input and the hidden state from the previous time step.

Input gate. The primary function of the input gate is to determine what information needs to be retained in the neuron based on the current input $x_{t}$. First, a candidate cell state at time *t* is generated using a tanh (the hyperbolic tangent activation function) layer, denoted as ${\tilde{C}}_{t}$. This candidate state is then combined with the forget gate’s output to update the cell state, resulting in a new cell state $C_{t}$. The specific computation process for the input gate is as follows:(22)$$i_{t}=\sigma (W_{i}\cdot [h_{t-1},x_{t}]+b_{i})$$(23)$${\tilde{C}}_{t}=\tanh(W_{C}\cdot [h_{t-1},x_{t}]+b_{C})$$(24)$$C_{t}=f_{t}*C_{t-1}+i_{t}*{\tilde{C}}_{t}$$
where $i_{t}$ represents the output state of the input gate at time *t*, determining the impact of the input $x_{t}$ on updating the cell state $C_{t}$; $W_{i}$ and $b_{i}$ denote the weight and bias terms of the input gate, respectively; $W_{c}$ and $b_{c}$ represent the weight matrix and bias for generating the candidate memory cell ${\tilde{C}}_{t}$; and ∗ denotes element-wise multiplication.

Output gate. The output gate extracts and outputs key information from the current cell state. First, a σ layer decides which neuron states should be output. These neuron states are then processed by a tanh layer and multiplied by the output of σ to produce the final output value $h_{t}$, which also serves as the hidden layer input for the subsequent time step. The calculation process for the output gate is as follows:(25)$$o_{t}=\sigma (W_{o}\cdot [h_{t-1},x_{t}]+b_{o})$$
where $o_{t}$ represents the output state of the output gate at time *t*; $W_{o}$ and $b_{o}$ are the weight matrix and bias terms for the output gate, respectively.

Due to these characteristics, LSTM is better suited for capturing dependencies in time series data, which is especially important for data with strong temporal associations. In this paper, combining LSTM with one-dimensional convolution allows for fully leveraging the convolution’s ability to extract local features, while LSTM captures the temporal dependencies in these severely coupled data.

## 4. Gravity Gradient Noise Signal Simulation and Compensation

This paper comprehensively considers various non-ideal factors in the error transmission mechanism in order to simulate the gravity gradiometer output signal. The parameters associated with these non-ideal factors (model parameters) are determined by selecting values at random from a specified range. Table 1 lists the specific values and units of the parameters, with ppm representing one part per million.

Based on the defined model parameters, the following input parameters are established: The distance *R* between the center of mass of the accelerometer and the center of the disk is set to 0.2 m. The sampling frequency of the rotating gravity gradiometer is set to 800 Hz, while the rotation frequency *f* of the GGI disk is set to 0.25 Hz. At *f* = 0.25 Hz, the primary influence on the measurement of the rotating gravity gradiometer is the component of each motion parameter below 1 Hz. The accelerations $a_{x}$, $a_{y}$, and $a_{z}$ in the three directions of the rotating gravity gradiometer are composed of randomly determined accelerations within the range of −0.3 to 0.3 $m/s^{2}$ and 100 components randomly distributed between 0.01 and 1 Hz. The amplitudes of these components are randomly determined between 0.006 and 0.010 $m/s^{2}$, and the phases are randomly determined. For $a_{z}$, the acceleration of 9.8 $m/s^{2}$ caused by gravity must be subtracted.

The angular velocities $\omega_{x}$, $\omega_{y}$, and $\omega_{z}$ in the three directions of the rotating gravity gradiometer are set to consist of determined angular velocities randomly determined within the range of −3 to 3 arcsec/s and 100 components evenly distributed from 0.01 to 1 Hz. The amplitudes of these components are randomly determined between 0.2 and 0.5 arcsec/s, and the phases are randomly determined. The above random numbers have a continuous uniform distribution, and quantities remain constant over time once determined. The angular accelerations of rotation in the three directions of the rotating gravity gradiometer are obtained using the first-order difference with respect to time for $\omega_{x}$, $\omega_{y}$, and $\omega_{z}$. Demodulation employs a low-pass filter with a rectangular window and a cutoff frequency of 0.075 Hz.

This paper simulates two sets of rotating gravity gradiometer output signals, denoted as signal A and signal B. Initially, signal A is used as the model’s training set, with signal B serving as the verification set. The compensation results are compared to the traditional deep learning method, MLP. Figure 4 shows the rotating gravity gradiometer’s outputs before and after compensation. The rotating gravity gradiometer output data, both before and after compensation, are demodulated [34]. The $\Gamma_{XX}-\Gamma_{YY}$ gradient component is demodulated with a sinusoidal doubling frequency, while the $\Gamma_{XY}$ gradient component is demodulated with a cosine doubling frequency. The $\Gamma_{XX}-\Gamma_{YY}$ and $\Gamma_{XY}$ gravity gradient components before and after compensation are obtained as shown in Figure 5 and Figure 6, respectively.

Subsequently, signal B is used as the training set for the model in this paper, while signal A serves as the verification set. The procedures are repeated, and the results before and after compensation are calculated using the rotating gravity gradiometer outputs before and after compensation, as shown in Figure 7. The $\Gamma_{XX}-\Gamma_{YY}$ and $\Gamma_{XY}$ gravity gradient components before and after compensation are shown in Figure 8 and Figure 9, respectively.

The outcomes of gravity gradient compensation experiments are evaluated using standard deviation (*STD*) and lift rate (*IR*). *STDs* are defined as follows:(26)$$STD=\sqrt{\frac{1}{n}\sum_{i=1}^{n}\left(\mu -\bar{\mu }\right)}$$
where n represents the sample size, μ is the signal value, and $\bar{\mu }$ is the mean.

IR is defined as follows:
(27)$$IR=\frac{STD_{u}}{STD_{c}}$$
where $STD_{u}$ is the uncompensated signal values and $STD_{c}$ is the compensated signal values.

The compensation result diagram and Table 2 of the gravity gradient component of the simulated data indicate that the deep learning method CNN-LSTM used in this paper outperforms the traditional MLP method, with a lower *STD* value and a higher *IR* value, resulting in a significant compensation effect on gravity gradient signal noise.

Residual neural networks have shown excellent applications in various fields such as denoising [35], geophysical inversion [36], and error compensation [37]. To further demonstrate the effectiveness of the method in this paper, we conducted a comparative analysis with a residual neural network, Res-BP. The comparison results are presented in Figure 10, Figure 11, Figure 12, Figure 13, Figure 14 and Figure 15.

The comparison results of Signal B:

The comparison results of Signal A:

From the results in Figure 10, Figure 11, Figure 12, Figure 13, Figure 14 and Figure 15 and Table 3, we can see that the compensation results of CNN-LSTM are better than Res-BP, demonstrating better performance in gravity gradient compensation of the simulated data.

## 5. Measured Gravity Gradient Data Compensation

This paper conducted a flight experiment in the airborne gravity gradient test area (Figure 16), which is located in the southern section of the Zhangguangcai Ridge, west of Mudanjiang City in Heilongjiang Province.

To ensure that the raw output of the gravity gradiometer only reflects dynamic measurement noise, a suitable flight altitude must be selected. At high altitudes, shallow surface or subsurface local targets have a negligible effect on the gravity gradient field. Over small areas, the gradient field appears constant, and it is reasonable to assume that gravity gradient data obtained at this altitude exclude gravity gradient signals caused by geological bodies. To achieve this, airborne gravity data for the area were collected. To establish a gravity gradient reference field, an upward continuation was performed using ground-measured data. Gravity anomalies on an observation plane at an altitude of 3600 m were calculated using arbitrary surface continuation theory, beginning with observed free-air gravity anomalies on the surface near the flight test area. Subsequently, frequency domain component transformation was then used to obtain gravity gradient anomalies $\Gamma_{XX}-\Gamma_{YY}$ and $\Gamma_{XY}$ at this altitude. Figure 17 shows that gravity gradient component anomalies $\Gamma_{XX}-\Gamma_{YY}$ and $\Gamma_{XY}$ within the black-framed area vary within 10 E, making this region suitable for high-altitude measurements at 3600 m. The actual flight trajectory is shown in Figure 18.

In the actual measurements, this paper uses a Y12 aircraft from China Flying Dragon General Aviation Co., Ltd. (Harbin City, Heilongjiang Province, China.), which is equipped with a gravity gradient acquisition system (see Figure 19). The system used in this study is a rotational gravity gradient measurement system, primarily composed of high-precision quartz accelerometers, weak signal processing circuits, a rotational mechanism, synchronization circuits, a power supply system, data acquisition and storage circuits, and electronic circuits. Among these components, the high-precision quartz accelerometer serves as the core sensor of the gravity gradient measurement system, primarily measuring acceleration sensitive to the test mass [38]. The aircraft repeatedly flew in a line at 3600 m altitude, collecting motion parameters such as accelerations, angular velocities, and gravity gradiometer output signals. These are combined with gravity gradiometer output data to create a deep learning sample set and carry out the compensation process.

Figure 20 shows the results before and after the compensation of the original data. The output data of the gravity gradiometer, both before and after compensation, are demodulated. Figure 21 and Figure 22 show the $\Gamma_{XX}-\Gamma_{YY}$ and $\Gamma_{XY}$ gravity gradient components before and after compensation, respectively.

Figure 21 and Figure 22 show how the deep learning approach using the CNN-LSTM effectively suppresses the dynamic noise of the gravity gradient. The compensation effect of this method outperforms that of the MLP neural networks.

Table 4 provides a summary of the final experimental results. The results demonstrate that the post-compensation method described in this paper effectively reduces noise in the actual data. When compared to traditional deep learning methods such as MLP, the improvement rate is significant, increasing the accuracy of gravity gradient compensation. For $\Gamma_{XX}-\Gamma_{YY}$, the *STD* value before compensation was 867.2312; after MLP compensation, it was 53.8435, and after CNN-LSTM compensation, it was 37.9429. The *IR* values are 16.1065 and 22.8562, respectively. For $\Gamma_{XY}$, the *STD* value before compensation was 457.4597; after MLP compensation, it was 24.4546, and after CNN-LSTM compensation, it was 15.7630. The *IR* values are 18.7065 and 29.0211, respectively.

Similarly, in the measured data, we also added a comparison with the Res-BP, and the results are shown below.

In the measured data, the CNN-LSTM still demonstrates better performance compared to Res-BP from the results shown in Figure 23, Figure 24 and Figure 25 and Table 5. This highlights the effectiveness of the method proposed in this paper.

## 6. Discussion

This study introduces a CNN-LSTM neural network compensation method for noise in gravity gradient dynamic measurements. The compensation results demonstrate the effectiveness of the network in gravity gradient compensation. In the simulated data, we considered the coupling between external motion parameters and instrument errors. However, in actual airborne gravity gradient measurements, the factors influencing the measurements are more complex, such as changes in aircraft fuel, external temperature fluctuations, atmospheric pressure variations, and so on. Because our compensation model does not take these factors into account, this makes the performance of the compensation model on measured data less perfect than on simulated data. In terms of compensation efficiency, the network used in this study is relatively complex, and the training process requires considerable time. In future research, we will try to improve compensation efficiency and focus on considering more factors affecting airborne gravity gradient measurements to enhance the accuracy of compensation.

## 7. Conclusions

In the process of dynamic airborne gravity gradient measurement, the primary source of noise in the gradiometer output is the coupling excitation between external parameters such as linear and angular motions and various non-ideal factors within the system. These interactions have complex, implicit, nonlinear relationships and temporal dependencies, and using traditional linear regression methods for post-error compensation has certain limitations. This paper considers the characteristics of dynamic noise in airborne gravity gradients. To address issues such as complex data relationships and strong coupling, it combines convolutional neural networks and long short-term memory neural networks to enhance the model’s feature extraction capability, effectively capturing the temporal dependency information between dynamic parameters. This approach improves the model’s feature extraction capabilities by effectively capturing temporal dependencies between dynamic parameters, thereby improving the model’s fitting capability in airborne gravity gradient compensation. In both the data simulated experiment and the application of actual airborne gravity gradient measurement data, the compensation results demonstrate effective suppression of dynamic measurement noise, significantly improving gravity gradient compensation precision. This validates the practicality and effectiveness of the proposed method for processing measured data.

## Figures and Tables

**Figure 1 sensors-25-00421-f001:**
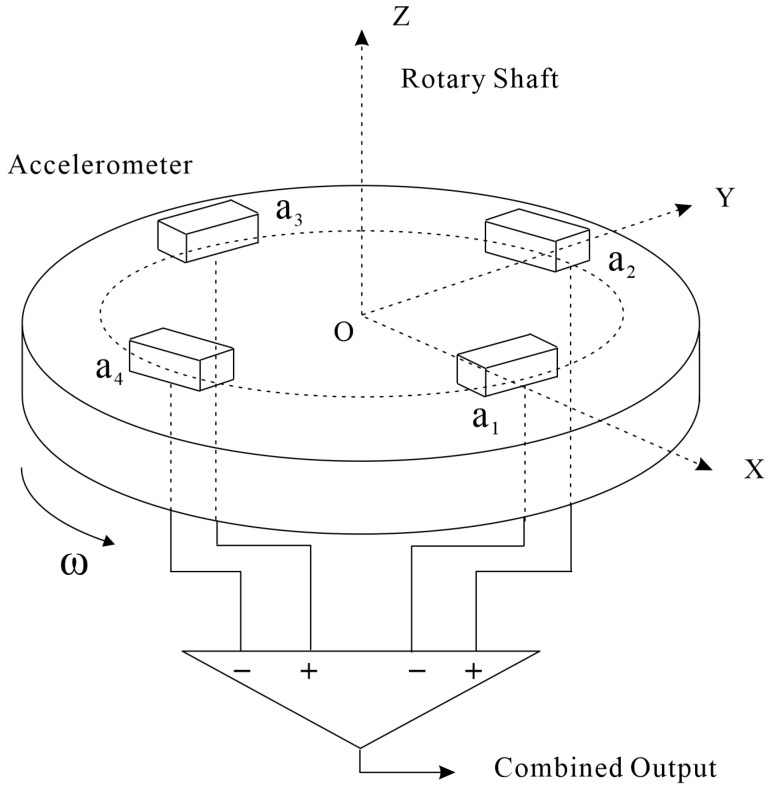
Diagram of the working principle of the rotary accelerometer gravity gradiometer.

**Figure 2 sensors-25-00421-f002:**
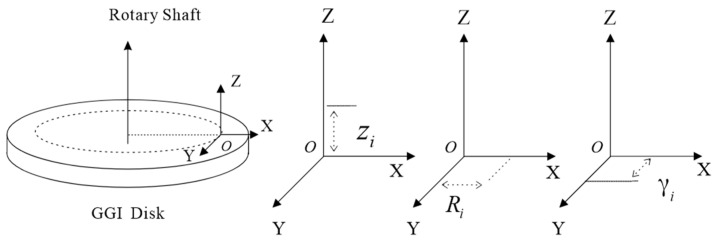
Schematic diagram of the accelerometer position mounting error.

**Figure 3 sensors-25-00421-f003:**
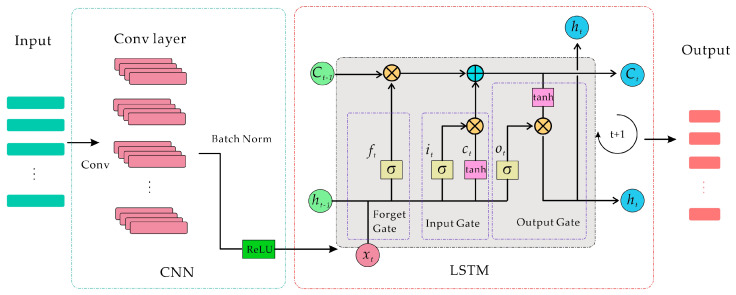
Neural network’s model diagram.

**Figure 4 sensors-25-00421-f004:**
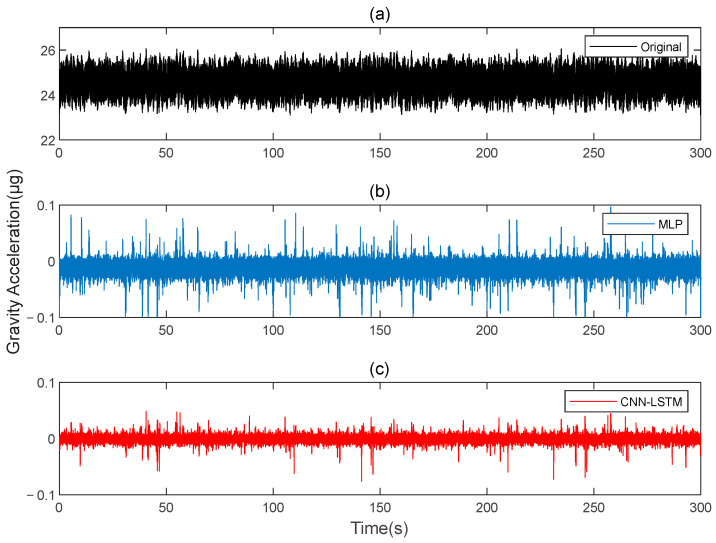
The gravity acceleration in Signal B: (**a**) original; (**b**) compensation results of MLP; (**c**) compensation results of CNN-LSTM.

**Figure 5 sensors-25-00421-f005:**
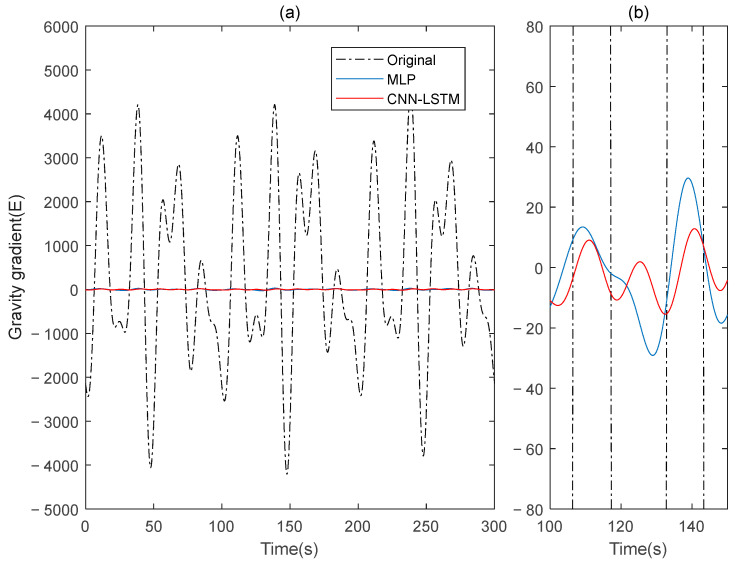
The $\Gamma_{XX}-\Gamma_{YY}$ gravity gradient compensated by MLP and CNN-LSTM in Signal B: (**a**) all; (**b**) part.

**Figure 6 sensors-25-00421-f006:**
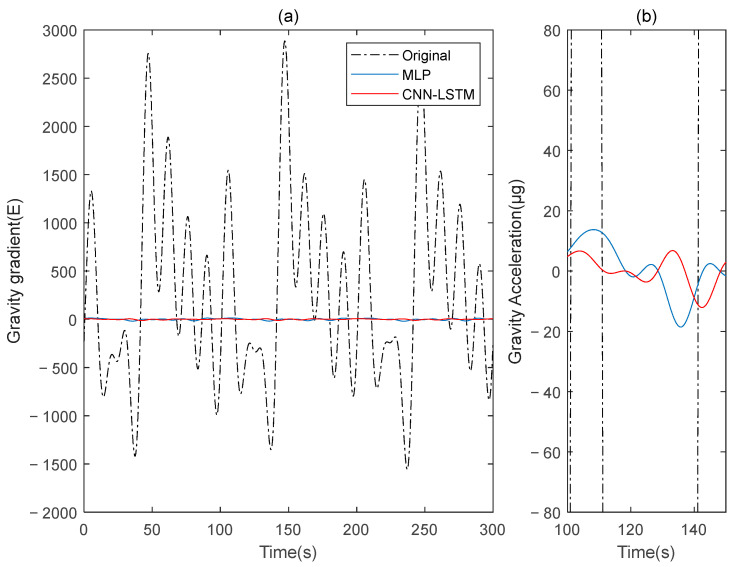
The $\Gamma_{XY}$ gravity gradient component compensated by MLP and CNN-LSTM in Signal B: (**a**) all; (**b**) part.

**Figure 7 sensors-25-00421-f007:**
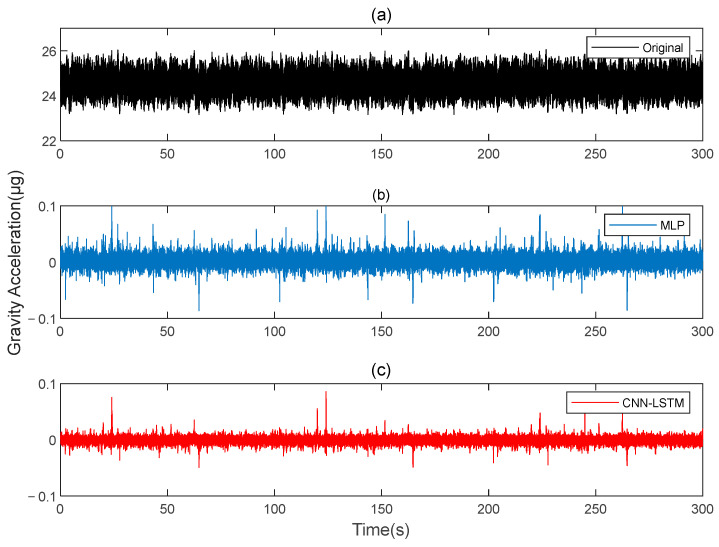
The gravity acceleration in Signal A: (**a**) original; (**b**) compensation results of the MLP; (**c**) compensation results of the CNN-LSTM.

**Figure 8 sensors-25-00421-f008:**
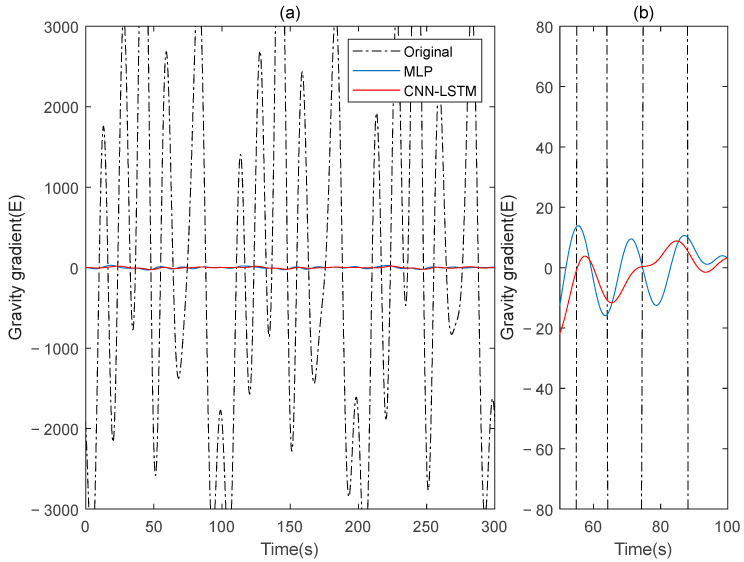
The $\Gamma_{XX}-\Gamma_{YY}$ gravity gradient component compensated by MLP and CNN-LSTM in Signal A: (**a**) all; (**b**) part.

**Figure 9 sensors-25-00421-f009:**
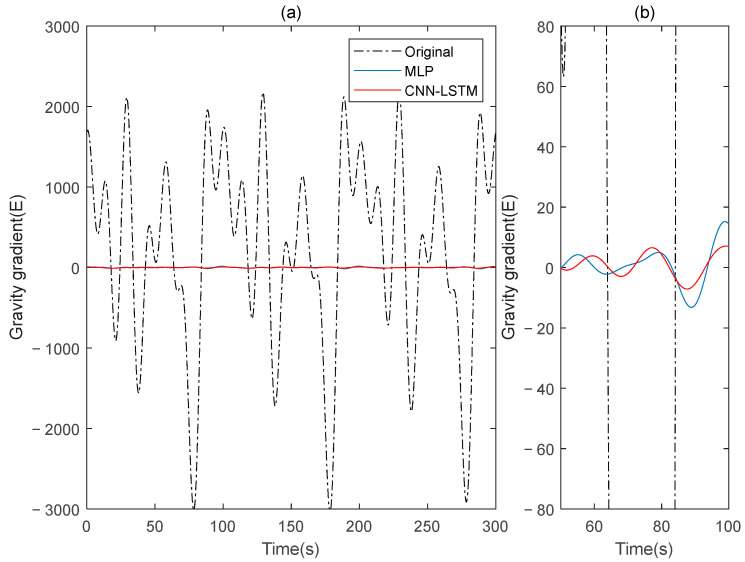
The $\Gamma_{XY}$ gravity gradient component compensated by MLP and CNN-LSTM in Signal A: (**a**) all; (**b**) part.

**Figure 10 sensors-25-00421-f010:**
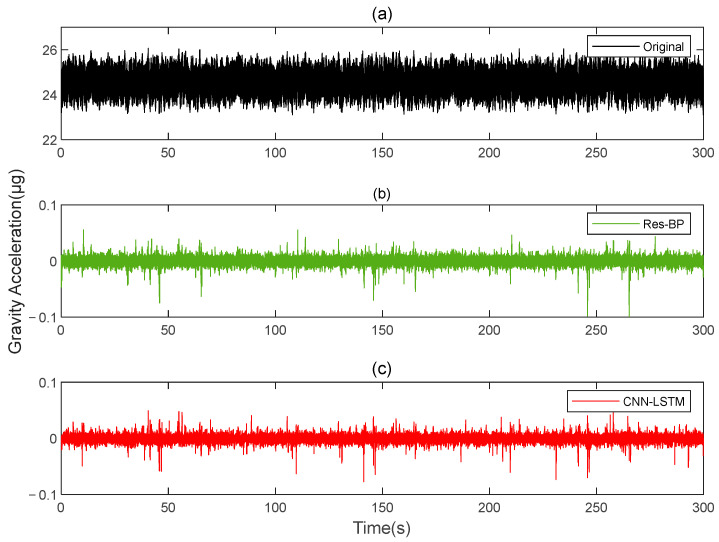
The gravity acceleration in Signal B: (**a**) original; (**b**) compensation results of Res-BP; (**c**) compensation results of CNN-LSTM.

**Figure 11 sensors-25-00421-f011:**
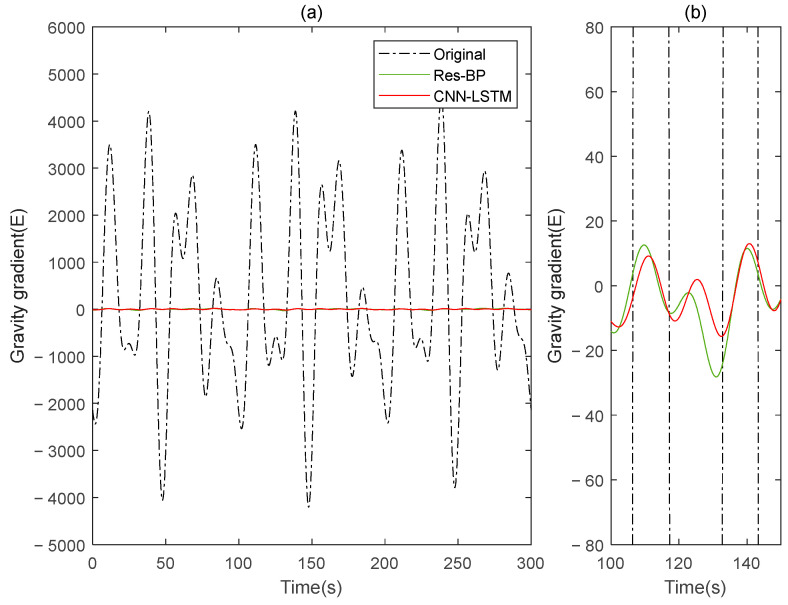
The $\Gamma_{XX}-\Gamma_{YY}$ gravity gradient component compensated by Res-BP and CNN-LSTM in Signal B: (**a**) all; (**b**) part.

**Figure 12 sensors-25-00421-f012:**
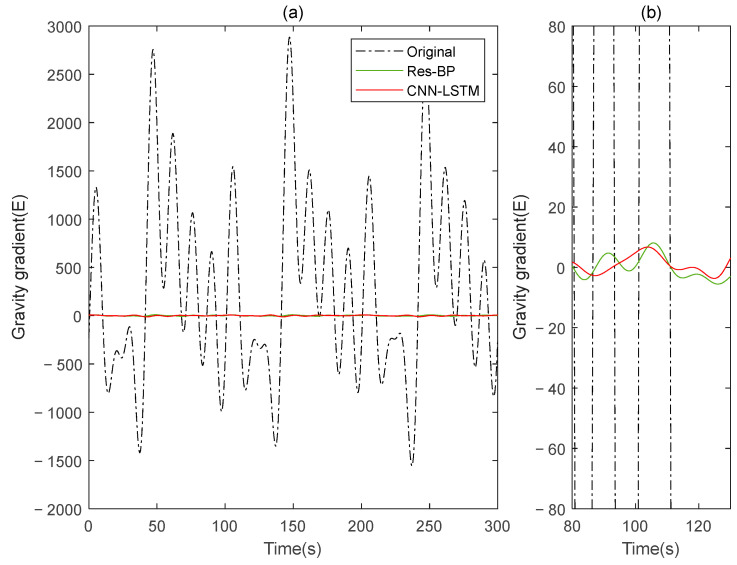
The $\Gamma_{XY}$ gravity gradient component compensated by Res-BP and CNN-LSTM in Signal B: (**a**) all; (**b**) part.

**Figure 13 sensors-25-00421-f013:**
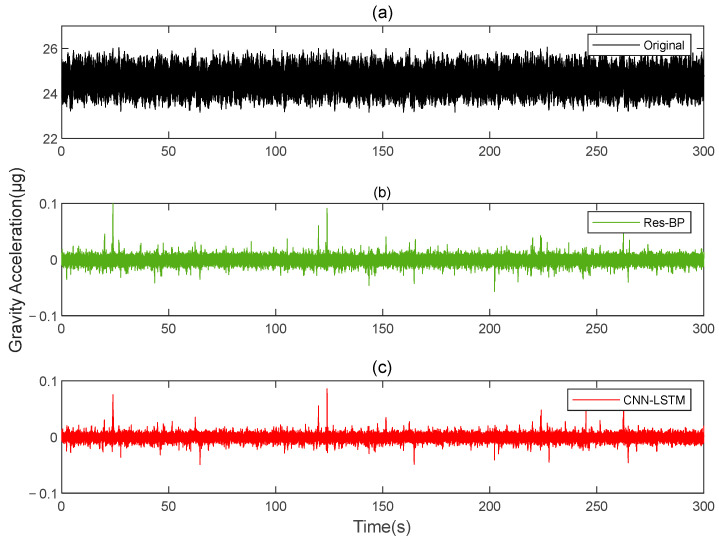
The gravity acceleration in Signal A: (**a**) original; (**b**) compensation results of the Res-BP; (**c**) compensation results of the CNN-LSTM.

**Figure 14 sensors-25-00421-f014:**
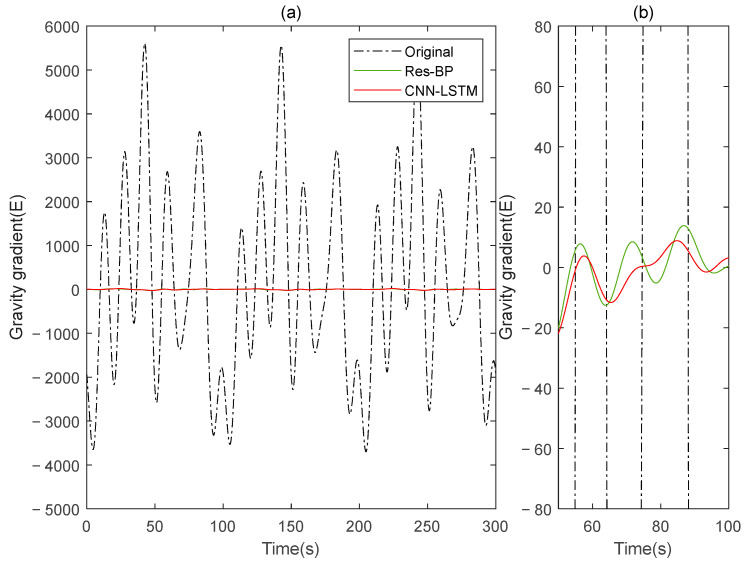
The $\Gamma_{XX}-\Gamma_{YY}$ gravity gradient component compensated by Res-BP and CNN-LSTM in Signal A: (**a**) all; (**b**) part.

**Figure 15 sensors-25-00421-f015:**
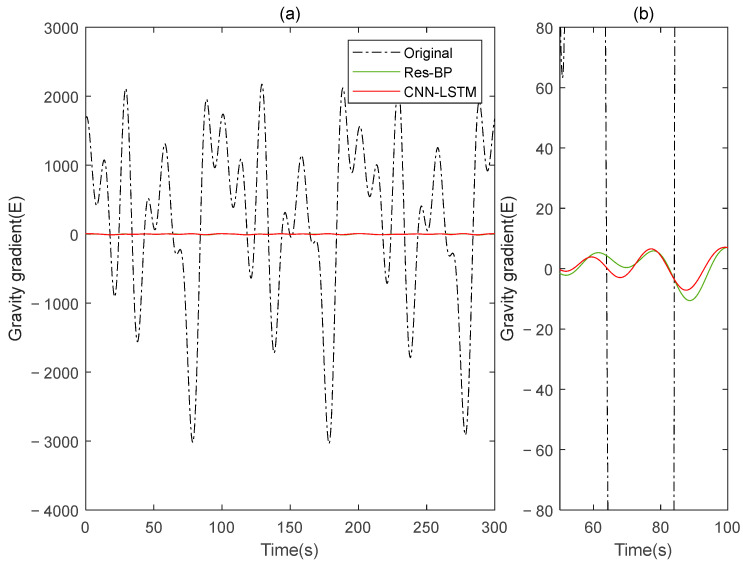
The $\Gamma_{XY}$ gravity gradient componentcompensated by Res-BP and CNN-LSTM in Signal A: (**a**) all; (**b**) part.

**Figure 16 sensors-25-00421-f016:**
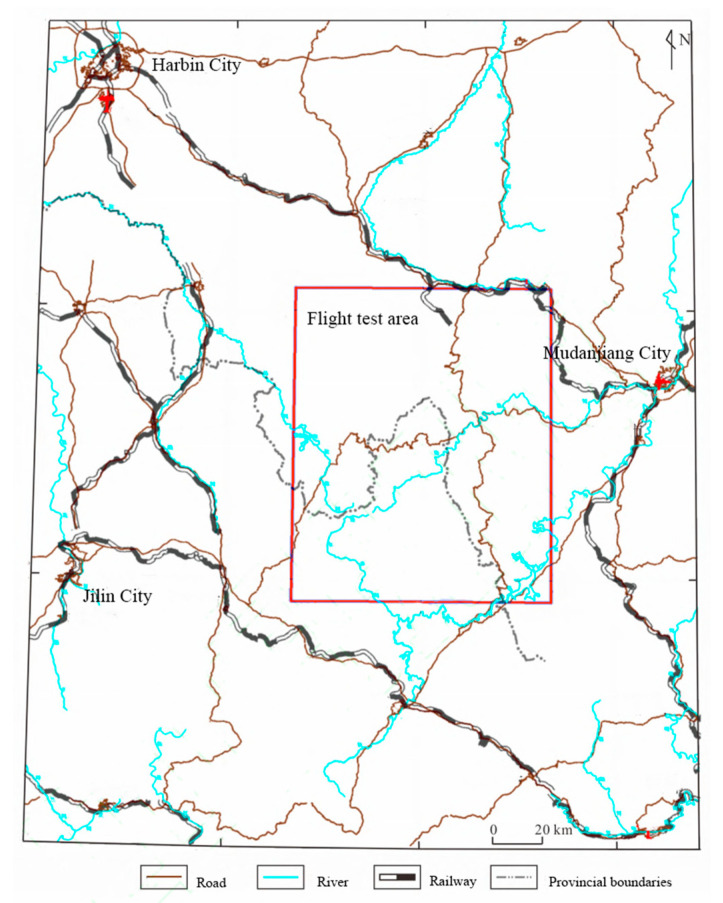
Schematic diagram of the extent of the flight test area of airborne gravity gradient.

**Figure 17 sensors-25-00421-f017:**
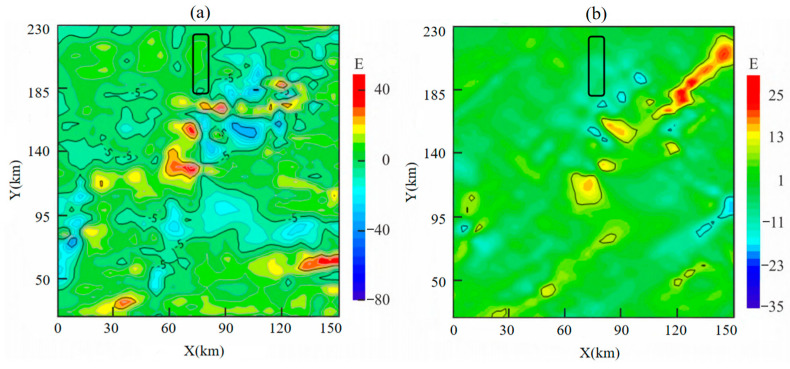
Gravity gradient reference field of measured data upward continuation to 3600 m: (**a**) $\Gamma_{XX}-\Gamma_{YY}$; (**b**) $\Gamma_{XY}$.

**Figure 18 sensors-25-00421-f018:**
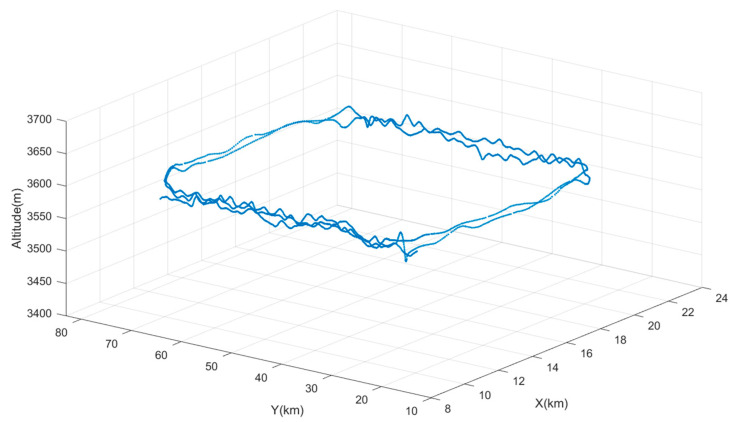
Actual flight trajectory.

**Figure 19 sensors-25-00421-f019:**
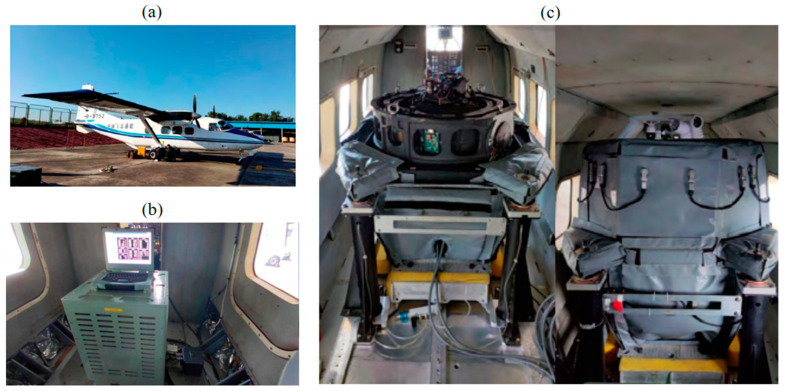
Gravity gradient acquisition system: (**a**) fixed-wing manned aircraft; (**b**) signal collection device; (**c**) the rotating gravity gradiometer.

**Figure 20 sensors-25-00421-f020:**
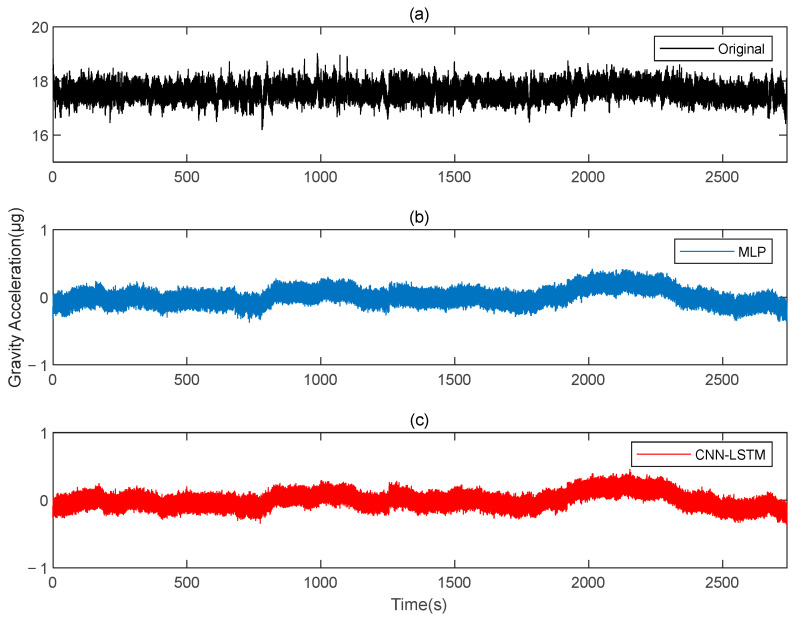
The gravity acceleration in measured signal: (**a**) original; (**b**) compensation results of the MLP; (**c**) compensation results of the CNN-LSTM.

**Figure 21 sensors-25-00421-f021:**
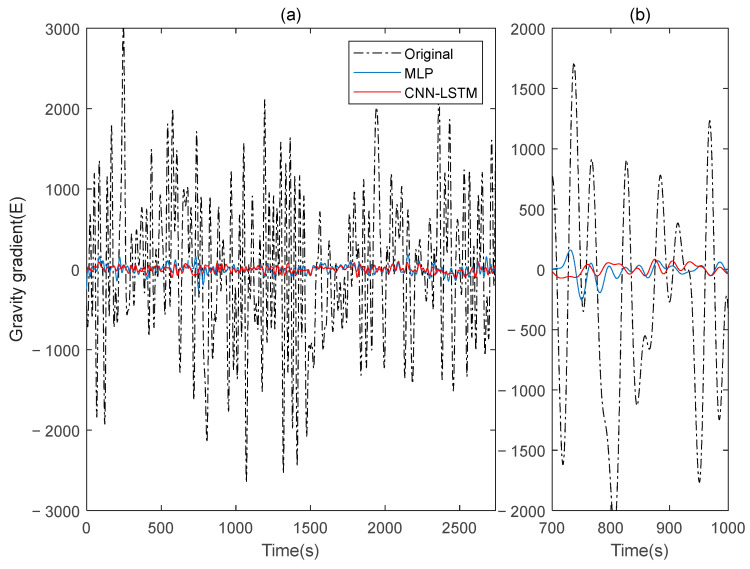
The $\Gamma_{XX}-\Gamma_{YY}$ gravity gradient component compensated by MLP and CNN-LSTM in measured signal: (**a**) all; (**b**) part.

**Figure 22 sensors-25-00421-f022:**
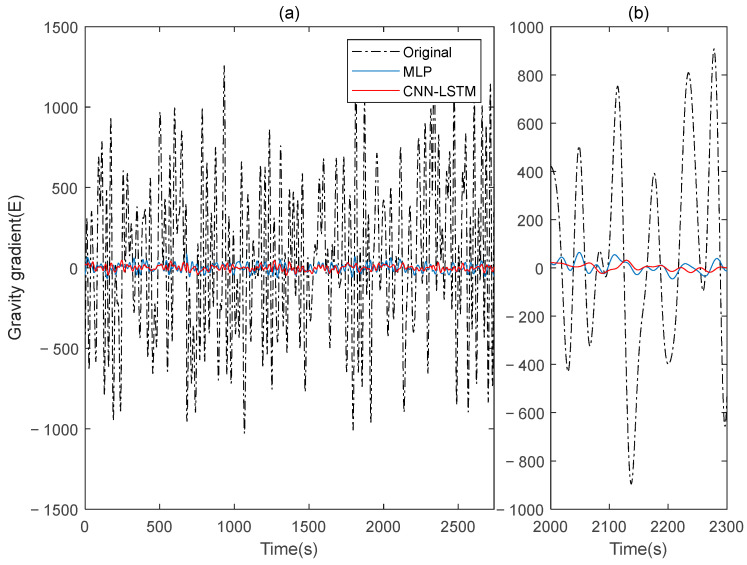
The $\Gamma_{XY}$ gravity gradient component compensated by MLP and CNN-LSTM in measured signal: (**a**) all; (**b**) part.

**Figure 23 sensors-25-00421-f023:**
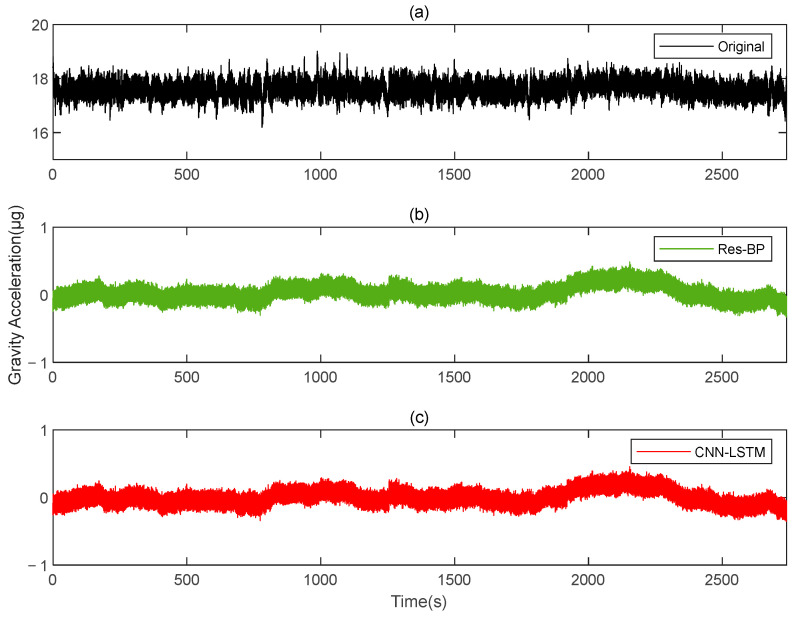
The gravity acceleration in measured signal: (**a**) original; (**b**) compensation results of the Res-BP; (**c**) compensation results of the CNN-LSTM.

**Figure 24 sensors-25-00421-f024:**
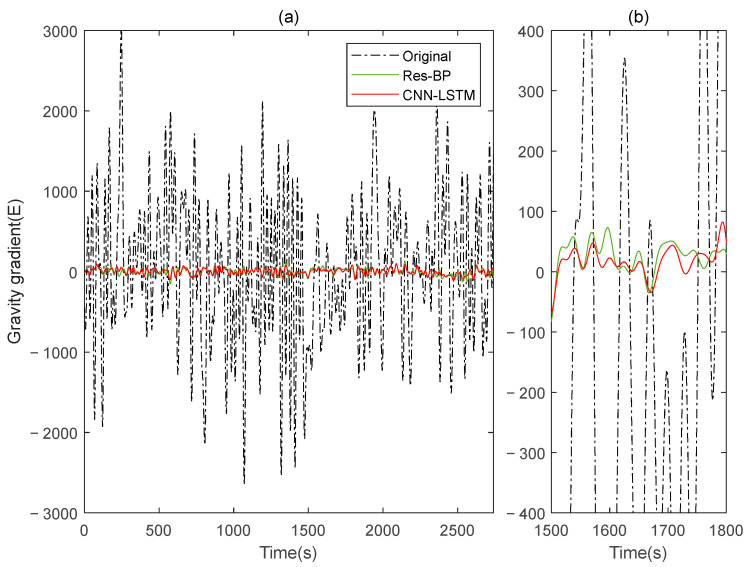
The $\Gamma_{XX}-\Gamma_{YY}$ gravity gradient component compensated by Res-BP and CNN-LSTM in measured signal: (**a**) all; (**b**) part.

**Figure 25 sensors-25-00421-f025:**
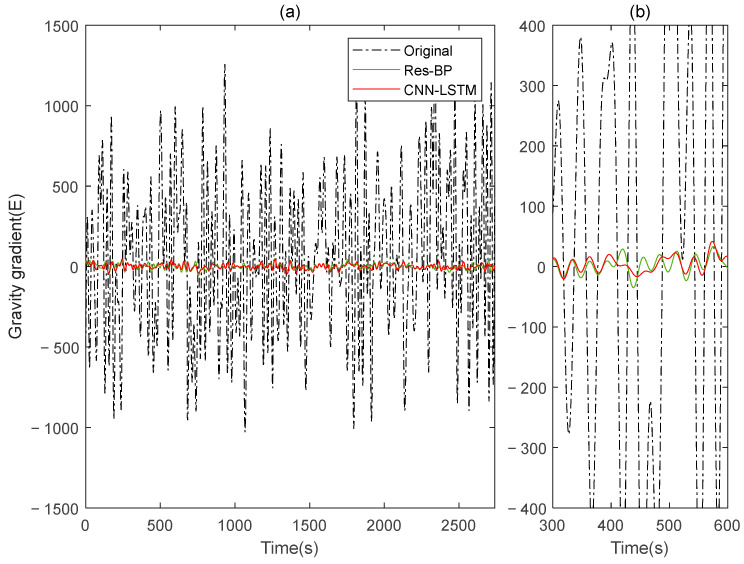
The $\Gamma_{XY}$ gravity gradient component compensated by Res-BP and CNN-LSTM in measured signal: (**a**) all; (**b**) part.

**Table 1 sensors-25-00421-t001:** Simulated model parameters.

Parameters	Units	Accelerometer			
		1	2	3	4
$z_{i}$	μm	46.27	36.40	−23.93	9.17
$R_{i}$		−9.36	−41.24	−4.11	46.91
$\gamma_{i}$		7.57	−36.80	23.04	−13.37
$\alpha_{Ii}$	arcsec	0.85	−1.07	−4.24	−2.22
$\alpha_{Oi}$		−2.57	−4.23	1.04	0.65
$\alpha_{Pi}$		4.32	2.66	4.45	1.23
$K_{IPi}$	$\mu g/g^{2}$	0.12	0.35	0.19	−0.55
$K_{IOi}$		0.32	−0.76	0.62	−0.74
$K_{OPi}$		−0.35	−0.55	0.73	0.44
$K_{Ii}^{\prime }$		0.62	−0.50	−0.13	0.52
$K_{Oi}^{\prime }$		0.84	0.28	0.68	−0.22
$K_{Pi}^{\prime }$		−0.37	0.54	0.11	−0.31
$K_{Oi}$	μg	−0.32	0.58	−0.79	0.45
$\frac{k_{i}}{k}-1$	ppm	−3.34	1.13	−1.36	−2.64

**Table 2 sensors-25-00421-t002:** The compensation results of MLP and CNN-LSTM in simulated data.

Data	Data	Method	${STD}_{u}$	${STD}_{c}$	*IR*
Signal A	$\Gamma_{XX}-\Gamma_{YY}$(E)	MLP	2197.7776	11.6645	188.4159
		CNN-LSTM		8.5535	256.9448
	$\Gamma_{XY}$(E)	MLP	1198.2159	5.5003	217.8456
		CNN-LSTM		4.3989	272.3928
Signal B	$\Gamma_{XX}-\Gamma_{YY}$(E)	MLP	1904.3852	13.2153	144.1056
		CNN-LSTM		8.0635	236.1725
	$\Gamma_{XY}$(E)	MLP	941.1884	8.4938	110.8089
		CNN-LSTM		4.3225	217.7421

**Table 3 sensors-25-00421-t003:** The compensation results of Res-BP and CNN-LSTM in simulated data.

Data	Data	Method	${STD}_{u}$	${STD}_{c}$	*IR*
Signal A	$\Gamma_{XX}-\Gamma_{YY}$(E)	Res-BP	2197.7776	8.9956	244.3177
		CNN-LSTM		8.5535	256.9448
	$\Gamma_{XY}$(E)	Res-BP	1198.2159	4.4896	266.8845
		CNN-LSTM		4.3989	272.3928
Signal B	$\Gamma_{XX}-\Gamma_{YY}$(E)	Res-BP	1904.3852	9.9179	192.0150
		CNN-LSTM		8.0635	236.1725
	$\Gamma_{XY}$(E)	Res-BP	941.1884	4.3548	216.1279
		CNN-LSTM		4.3225	217.7421

**Table 4 sensors-25-00421-t004:** Comparison of compensation results of the MLP and CNN-LSTM in measured data.

Data	Method	${STD}_{u}$	${STD}_{c}$	IR
$\Gamma_{XX}-\Gamma_{YY}$(E)	MLP	867.2312	53.8435	16.1065
	CNN-LSTM		37.9429	22.8562
$\Gamma_{XY}$(E)	MLP	457.4597	24.4546	18.7065
	CNN-LSTM		15.7630	29.0211

**Table 5 sensors-25-00421-t005:** Comparison of compensation results of the Res-BP and CNN-LSTM in measured data.

Data	Method	${STD}_{u}$	${STD}_{c}$	IR
$\Gamma_{XX}-\Gamma_{YY}$(E)	Res-BP	867.2312	40.2761	21.5322
	CNN-LSTM		37.9429	22.8562
$\Gamma_{XY}$(E)	Res-BP	457.4597	16.4987	27.7271
	CNN-LSTM		15.7630	29.0211

## Data Availability

Data available on request due to restrictions.

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
