# Peer review of "An Airborne Gravity Gradient Compensation Method Based on Convolutional and Long Short-Term Memory Neural Networks"

_sensors, 2025, doi:10.3390/s25020421_

Round 1
Reviewer 1 Report
Comments and Suggestions for Authors
The paper presents a novel post-error compensation algorithm that utilizes Convolutional Neural Networks (CNN) combined with Long Short-Term Memory (LSTM) networks to enhance the accuracy of airborne gravity gradient measurements. This method effectively addresses the challenges posed by the dynamic environment of aircraft motion and attitude changes, which can significantly impact measurement results. By leveraging the feature extraction capabilities of convolution and the temporal dependency handling of LSTM, the proposed model demonstrates superior performance compared to traditional Multi-Layer Perceptrons (MLP) in compensating for measurement errors in both simulated and real-world data from Heilongjiang Province
.
To improve the paper, the following suggestions can be considered: First, the authors could include a more detailed comparison of their method with other state-of-the-art compensation algorithms to highlight its advantages more clearly. Second, incorporating a larger and more diverse dataset for training and testing could enhance the model's robustness and generalizability across different airborne conditions. Third, the paper could benefit from a more thorough discussion on the limitations of the proposed method, including potential scenarios where it may underperform. Fourth, providing a more comprehensive analysis of the computational efficiency and resource requirements of the CNN-LSTM model would be valuable for practitioners considering its implementation. In addition, future work could explore the integration of additional data sources, such as environmental factors or sensor fusion techniques, to further improve the accuracy of gravity gradient measurements and enhance the model's predictive capabilities. The authors are invited to include some recent references, especially some references related to Deep Convolutional Neural Networks. For instance, the authors may include the following interesting references (and others):
a. https://www.mdpi.com/2073-431X/12/8/151
b. https://www.taylorfrancis.com/chapters/edit/10.1201/9781003393030-10/learning-modeling-technique-convolution-neural-networks-online-education-fahad-alahmari-arshi-naim-hamed-alqahtani
Reviewer 2 Report
Comments and Suggestions for Authors
This paper deals with an improved post-error correction compensation algorithm to reduce the impact of dynamic environment on the accuracy of gravity gradient measurements. In the first chapter is a thorough introduction to the field of gravity gradient measurements, while the second chapter deals with the measurement equation and error transmission mechanism of rotating gravity gradiometer. In the third chapter the Authors describe in detail their new correction method and in the fourth chapter (wrongly identified again as third) they test it on simulated data showing a good increase in performance. In the final chapter the new method is applied to real data taken on field, showing a significant but somehow less relevant increase in sensitivity.
In general, the paper is well written and clear in explaining the new method proposed by the Authors. However there are some remarks that I would like to propose, in order to improve its effectiveness.
- On page 2 line 76: the sentence "resulting in wave-particle duality and the ability..." could be misleading: of course the wave particle duality is a general feature of particles and does not come out of the very low temperature. I propose to write simply: "resulting in the ability..."
- On page 7 line 260: in the description of the complex error analysis of the rotating gravity gradiometer there is some phrasing missing. Please revise these sentences.
- On page 8 line 296: "Because of the limited mechanical implementation capacity": this statement is too generic, and I would like to have more details on what these limits are. Also, I would appreciate some reference or short description regarding the accelerometers being used.
- On page 9 line 351: Some sentences are repeated twice. Please correct.
- On page 10, Fig.3: in the natural networks model diagram, it would be useful for the reader to highlite the different gates described in the text, e.g. the Forget Gate, etc.
- On page 10 line 397: the tanh layer is first quoted here, but described later, in line 407. This of course should be changed.
- The last chapter shows the results of testing the new method in the field: here the results are not so relevant as for the simulated data in the previous chapter. I think that the quality of the paper will be improved by some discussion of possible causes of this difference, and of the actions that the Authors plan in order to overcome this issue.
- Finally in Table 3 on page 20, but also in the text, values of evaluated parameters with 6 digits precision are reported. Perhaps here the number of digits is not related to the precision of the measured data: I invite the Authors to verify this point.
As a concluding remark, I find this paper well written and convincing in presenting the new technique proposed by the Authors, and I look forward to having a new versions of the paper where the points that I raised above were taken into account before its publication in Sensors.
Round 2
Reviewer 1 Report
Comments and Suggestions for Authors
The authors considered my comments and suggestions. Good luck.